# Thermal Decomposition Kinetics of Basalt Fiber-Reinforced Wood Polymer Composites

**DOI:** 10.3390/polym12102283

**Published:** 2020-10-05

**Authors:** Xian Zhang, Runzhou Huang

**Affiliations:** 1Co-Innovation Center of Efficient Processing and Utilization of Forest Products, College of Materials Science and Engineering, Nanjing Forestry University, Nanjing 210037, China; shinezhang17@163.com; 2School of Renewable Natural Resources, Louisiana State University Agricultural Center, Baton Rouge, LA 70803, USA

**Keywords:** basalt fiber, thermal decomposition, TGA, kinetics, apparent activation energy

## Abstract

Thermogravimetric analysis (TGA) was used for the observation of the pyrolysis kinetics characteristics of high density polyethylene (HDPE)-based composites enhanced by a variety of basalt fibers (BFs) and wood flour (WF). The improved Coats-Redfern (C-R), Flynn-Wall-Ozawa (F-W-O), Friedman, and Kissinger methods were utilized to ascertain the specific apparent activation energy (*E_a_*) of each component and composite material. The results indicate that BFs do not decompose under 800 °C, while the pyrolysis of WF and waste HDPE showed two significant weight loss zones (250–380 °C and 430–530 °C), relative to cellulose/hemicellulose and HDPE thermal degradation, respectively. The average *E_a_* of WF/BF/HDPE composites over the entire pyrolysis process obtained by the modified C-R method fluctuated in a range of 145–204 kJ/mol and increased with the BF content, which was higher than that of WPC (115–171 kJ/mol). The value of *E_a_* computed by the F-W-O method was significantly lower than that computed with the improved C-R method, which could validate the reliability of two methods by comparing with the literature. The Friedman and Kissinger methods were not applicable to this composite material reinforced by mixed fillers, so the obtained *E_a_* values were quite different from the previous two methods. The changes in *E_a_* showed that the addition of BFs could improve the average *E_a_* and further enhance the thermal stability and flame resistance of the composites.

## 1. Introduction

Over the past few years, wood-plastic composites (WPCs), which show advantages with respect to biodegradability, low cost, superb mechanical and thermal insulation performance, and availability, have been regarded as a novel green renewable material exhibiting great potential for structural building applications [1,2,3,4]. However, the increasing environmental concerns and market demands on natural fibers mean that reinforced polymer matrices in WPCs must be more environmentally friendly, cost-effective, and higher strength [5]. An array of studies have assessed the applicability of both organic and inorganic fibers in polymers. Among these fibers, mineral basalt fibers (BFs) have attracted intense research interest as a reinforcing material.

Conventional fiber materials include glass (GFs) and carbon fibers (CFs) that form popular fibers in polymer composites [6]. However, the use of these fiber composites is hindered by serious drawbacks including high energy costs for production, inhalation health risks, non-recyclability, and lack of biodegradability [7,8]. In this context, BFs, as a natural reinforcing material in polymers, provide a promising alternative to WPCs. The fibers consist of SiO_2_ and Al_2_O_3_ and are composed of natural ores via melting at ~1500 °C and have a significant number of capabilities including economical costs, high strength, excellent oxidation and corrosion resistance, are non-flammable, and show high insulative properties [9,10,11,12]. Furthermore, BFs have higher heat resistance and mechanical stability than both GFs and CFs [13,14,15]. Bashtannik et al. [16] successfully prepared polypropylene (PP) reinforced with BFs via extrusion and concluded that BF-reinforced polymers display excellent wear properties. Ozturkl et al. [17] showed that BF content initially intensified the resistance and peel strength of basalt/phenolic resin composites, which decreased when the BF content exceeded recommended levels, showing optimal properties at 48 wt.%. Wu et al. [18] discussed the thermal properties and morphologies of BF-loaded high density polyethylene (HDPE) and co-extruded WPCs with BF/HDPE composite shells. It was shown that BF, when present in the HDPE matrix, increased flexural expansion and increased dynamic and tensile modulus. However, compared with HDPE alone, BF/HDPE composites display a simplified linear coefficient of thermal expansion (LCTE).

Compared with other common plastics (such as PP and PE), HDPE, with a better low temperature resistance and relatively cheap price, has many mechanical property advantages, such as improved rigidity, hardness, and tensile strength [19]. The flexural and tensile properties of composite materials obtained by adding wood materials into polymers have been shown to be significantly improved, dimensional stability is maintained, and the environment will not be polluted during or after its useful lifetime [20,21,22]. WPCs have been favored by the vast majority of people, and the usage and output of WPCs are rapidly increasing every year. However, both the wood fiber and polymeric materials in WPC are flammable materials. Once exposed to fire, it is easy to cause further fires, and poses a serious threat to people’s lives, which restricts the application field of the material. Therefore, basalt fibers were selected to be added to WPC as a way to improve its thermal stability. Basalt products can be used over a wide temperature range (about −200~800 °C). Bazan et al. [23] added basalt fiber into poly (oxymethylene) and observed higher thermal and dimensional stability with very good strength and excellent stiffness for BF-reinforced composites. Ying et al. [24] reported the BFs had better strength retention than glass fibers at a relatively high temperature.

Thermogravimetric analysis (TGA) has extensive applications in the analysis of polymeric material composition, with the ability to ascertain thermal stability, decomposition reactions, and material kinetics. Kinetic analysis of pyrolysis reactions is important in understanding physiochemical changes during pyrolysis processing of materials. The thermal performance of WPCs is decided by a mix of factors related to heat and temperature, and the pyrolysis characteristics of composites have been evaluated [25,26,27]. Zhang et al. [28] fabricated BF-reinforced poly butylene succinate (PBS) composites using distinct molding methods and characterized how the heat resistance of the BF/PBS composites were enhanced with BF incorporation. Kim [29] demonstrated that the TGs of BF composites grew by ≥15 °C for epoxy-amine composites and ≥ 20% for epoxy-anhydride composites. BFs supplementation into the WPCs enhanced the performance of all composites assessed.

Herein, to systematically assess the thermal stability of WF/BF/HDPE composites, we investigated the influence of heating rates on pyrolysis through TGA. The modified Coats-Redfern (C-R), Flynn-Wall-Ozawa (F-W-O), Friedman, and Kissinger protocols were employed for the assessment of apparent activation energy (*E_a_*) and to perform thermal decomposition measurements of the composites to enhance our understanding of the reaction mechanisms and pyrolysis characteristics during thermogravimetric decomposition.

## 2. Methods

### 2.1. Study Agents/Fibers

BFs (diameter = 7 µm, length = 2.5 cm, density = 2.6 g/m^3^) were supplied by the US-headquartered company CETCO Oilfield Services (Broussard, LA, USA). Wood flour was obtained from American pine (particle size: 0.6–0.75 mm). Coupling agents including maleic anhydride grafted polypropylene (MAPE), Epolene^TM^ G2608 with a melt flow rate (MFR) of 6 to 10 g/10 min at 190 °C/2.16 kg, MF = 65,000 g/mol, and acid number 8 mg KOH/g were purchased from Eastman Chemical Co. (Kingsport, TN, USA) to enhance filler and plastic compatibility. HDPE (AD60-007) with an MFR = 0.7 g/10 min at 190 °C/2.16 kg and density = 0.96 g/cm^3^ was purchased from Exxon Mobil Chemicals (Houston, TX, USA).

### 2.2. Composite Sample Preparation

BF, WF, HDPE, and 2% MAPE (accounts for 2% of the total mass fraction) were melt compounded (Table 1) using a counter-rotating Brabender twin-screw extruder equipped with an intermesh at 40 rpm. Feeding zone temperatures of the extruders were set at 150 °C and the plasticizing and homogenizing zone temperatures were 175 °C. Extrudates were cooled and pelletized. Pellets were dried in the oven at 70 °C for 12 h before experimental analyses.

### 2.3. Test Conditions for Thermogravimetric Analysis

Thermogravimetric decomposition assessments were employed for the assessment of global mass loss on a TG209F3 analyzer (Netzsch Group, Selb, Germany). Experimental temperatures were set at 30 ± 3 °C to 800 °C at four heating rates for a period of 5 to 20 °C/min. Samples (6 mg) were taken and dried at 70 °C for 24 h before testing. Experiments were performed at room temperature (RT) under 99.5% nitrogen and 0.5% oxygen at a 30 mL/min flow rate.

### 2.4. Data Analysis

Kinetic studies were assessed using the fundamental rate equation:(1)$$d\alpha /dt=kf\left(\alpha \right)$$
where k: rate constant and *f(α)*: reaction model decided by the reaction mechanism.

Equation (1): rate of conversion; dα/dt: fixed temperature according to the loss of the reactant and rate constant.

Conversion rate (α):(2)$$\alpha =\left(W_{0}-W_{t}\right)/\left(W_{0}-W_{f}\right)$$
where *W_0_*, *W_t_* and *W_t_*: initial, time *t*, and final weights of samples, respectively.

*k* was defined using the Arrhenius equation:(3)$$k=Aexp\left(-E_{a}/RT\right)$$

Where *E_a_*: activation energy in kJ/mol, *R*: gas constant (8.3145 J/K mol), *A*: pre-exponential factor (min^−1^), and *T* absolute temperature (K).

Combining Equations (1) and (3):(4)$$d\alpha /dt=Aexp\left(-E_{a}/RT\right)f\left(\alpha \right)$$

Dynamic TGA is calculated through the introduction of the heating rate,  β=dT/dt, into Equations (4) and (5), for which the following holds:(5)$$d\alpha /dt=\left(A/\beta \right)exp\left(-E_{a}/RT\right)f\left(\alpha \right)$$

This equation represents the assessment of the kinetic parameters of the TGA data.

The F-W-O and improved C-R methods (Table 2) are frequently employed to measure the activation energy [30]. The iso-conversional F-W-O results in $-E_{a}/R$ from the slope up to lgβ  versus 1/T with a given rate of conversion. The improved C-R method uses $\ln\left(\beta /T_{P}^{2}\right)$ against 1/T assessed at a range of temperatures and heating rates from TG (DTG, derivative thermogravimetry) curves. Under the framework of the Friedman method, $\ln\left(d\alpha /dt\right)$ should be drawn in line with 1/T during a set of tests and the Kissinger method explicitly results in $-E_{a}/R$ of the term $\ln\left(\beta /T^{2}\right)$ according to $1/T_{P}$ with heating rates at the peak temperature, *T_P_*, of the DTG curve.

## 3. Results and Discussion

### 3.1. Thermal Decomposition Characteristics

The thermogravimetric decomposition process of raw materials at a given heating rate of 20 °C /min is shown in Figure 1. The thermal degradation of cellulosic materials occurring in the composite material processing process may influence the composite performance [30,31,32]. At a lower temperature (below 100 °C), BFs and waste HDPE almost did not decompose in the TG curves, and the residual mass was nearly 100%. Wood flour had a weight loss of ~7 % as a result of the loss of adsorbed water and crystal water from the fibers [33]. The residue of BFs, WF, and waste HDPE was 97.1%, 15.5%, and 3.9%, respectively, at the end of test, which were quite different. BFs had only 3% weight loss in the entire pyrolysis process, which was consistent with the literature [34], and the residual mass was the largest among the three raw materials, indicating that BF had a good high temperature resistance. At a given heating rate, the pyrolysis of wood mainly went through four stages with increasing temperature. The first stage is from 100 to 250 °C, where the weight loss of wood materials was very small, because only a portion of the cellulose was dehydrated at lower temperature to produce dehydrated cellulose, which was a slow process. The second stage was from 250 °C to 450 °C, which is the main stage of samples pyrolysis. In this range, hemicellulose and cellulose in the wood pyrolysis reaction produce bio oil, dehydrated cellulose further generated small molecular gas and coke, resulting in significant weight loss, and the weight loss rate reached a maximum value at about 350 ° C. The last area corresponded to the slow decomposition of lignin and the final residue, and some carbon and ash were generated [35,36,37]. The two weightless peaks in Figure 1 of waste HDPE, with complex components, were the decomposition of wood fiber and HDPE. The main structure of HDPE was C-C as the main chain, and its pyrolysis process is a free radical chain reaction. According to the TG-DTG curves of materials in Figure 1, it can be deduced that the second part was mainly the fracture of the C-C main chain in HDPE, which is the pyrolysis macromolecular polymers into small molecules. The weight loss rate of continuous BFs was the lowest relative to the others, because basalt fibers were mainly composed of SiO_2_ as the main component, Al_2_O_3_ as the secondary, and other oxide ceramic components, which leads to basalt fibers showing excellent high temperature resistance performance [38]. The initial decomposition temperature *T_0_* (obtained by the corresponding temperature when the weight loss percentage was 5%) of the three raw materials at the differential heating rates β, the peak temperature *T_P_* (the temperature at the maximum point on the DTG curve), WL (ratio of weight loss percentage relative to *T_0_* and *T_P_* is the marked sample, respectively), and residual amount are shown in Table 3. As the weight changes of BFs were so small that the mass loss rates were less than 5 % at 800 °C, the following studies on the decomposition and average *E_a_* of BFs were no longer conducted.

Table 3 presents the thermal decomposition parameters of the raw materials at the indicated heating rates. The heating rate of the raw material temperature lagged behind that of environmental temperatures, meaning the *T_0_* and *T_P_* increased. The temperatures when DTG reached the peaks of WF and waste HDPE at a heating rate of 5 °C/min were 342.0 and 447.5 °C, respectively. Once the heating rate approached 20 °C/min, the maximum *T_P_* of WF and waste HDPE appeared at 357.4 °C and 469.6 °C, respectively, which were higher than that at 5, 10, and 15 °C/min.

The thermogravimetric decomposition process of wood plastic composites and WPCs reinforced with varying amounts of BFs at a heating rate of 20 °C/min is shown in Figure 2. With reference to Figure 2, the general trend of TG and DTG curves for WPCs and WPCs reinforced with BFs was similar, and the two significant weight loss zones of the composites were in the 250–350 °C and 440 –540 °C ranges. The first weight loss stage of wood-plastic composites was mainly pyrolysis of the mixture of cellulose and hemicellulose in wood flour, and the second was the fracture of the C-C main chain in HDPE [35]. The coke produced by the pyrolysis of hemicellulose and cellulose covered the surface of HDPE and basalt fibers, which prevented further pyrolysis of the composite materials, and the salt minerals in basalt fibers also have some chemical reactions during this stage; for example, FeO was oxidized to Fe_2_O_3,_ which makes the pyrolysis of composites more difficult [23]. Thermal decomposition parameters, *T_0_* and *T_P_**,* and residual amounts of composites with different BF contents with varying heating rates are listed in Table 4. When the heating rate grew from 10 °C/min to 30 °C/min, *T_0_* of all composites tended to move towards a high temperature side, and the temperature relative to maximal weight loss rate also tended to increase, which was consistent with the view of Kim et al. [39]. This is because the outside-to-inside heat transfer of the particles takes a longer time at a higher temperature rise rate [40], which reduces the heat transfer efficiency, so the *T_P_* rises when the maximum weight loss rate is attained. In addition, there was no distinct difference in residual mass among the four heating rates observed, indicating that the change in heating rate made no obvious impact on residue. Compared with A2 (without loading BFs in composites), the *T_0_* and *T_P_* of composite materials with BF addition shifted to higher temperatures, and with a higher BF content, the *T_0_* and *T_P_* values were higher. In addition, the content of BF mainly affected the change in the residual amount of the thermal decomposition process of composites, and the residual levels of WF/BF/HDPE composites were significantly greater than those of WF/HDPE composites; furthermore, the composite residuals gradually increased with the increasing BF content, indicating the filling of BFs is beneficial in improving the thermal stability of composites.

Heating rates significantly impact the thermogravimetric process of materials (Table 3 and Table 4). With increasing heating rates, the thermogravimetric curves reach higher temperatures, causing the extent of the weight loss of materials to vary. Generally, it is considered that the difference in the degree of weight loss under different heating rates are due to the differences in the reaction process and secondary reactions of WPC at varying heating rates, resulting in the discrepancy concerning wood flour carbonization. In addition, with increasing heating rates, the main reaction ranges of the materials are widened.

The *T_0_* and *T_P_* values of all materials changed with the increased heating rates, as the actual temperature of the materials lagged behind the surrounding temperatures with the increased heating rates during testing. To exclude the effects of the heating rate, *T_0_* and *T_P_* were retrieved from the linear extrapolation β=0. Figure 3 shows the decomposition characteristic parameters with 6.7% BF content (B3). According to its *T_0_* and *T_P_* values under different heating rates, linear extrapolation methods were adopted to determine the initial *T_0_* and *T_P_* values [41]. Table 5 lists *T_0_*, *T_P_*, and residual mass of composites after linear fitting. It is shown that there is no significant change in *T_0_* and *T_P_* among wood plastic composite materials with different wood flour concentrations. Compared with wood plastic composites, the addition of BFs has little impact on the *T_0_* and *T_P_* of materials after eliminating the influences of heating rates, indicating that the BF content within the scope of the test had little influence on the material characteristics of the pyrolysis process temperature, but the addition of BF increased the residual mass of the composites.

### 3.2. Kinetics of WPC

Upon assessment of the pyrolysis kinetics of the materials, the activation energy (*E_a_*) tendencies of the materials were assessed. Figure 4 presents representative iso-conversional plots drawn by the modified C-R and F-W-O methods for waste HDPE and wood-plastic composites (A2), respectively. Assuming a reaction order of n = 1 according to the modified C-R method, the connectivity of $\ln\left(\beta /T^{2}\right)$ and 1/T at different conversion rates ranging between 0.1 and 0.9 is shown in Figure 4, and the kinetic parameters calculated with varying heating rates are listed in Table 6. The fitting lines of different conversion rates were close to parallel, suggesting that they had an approximate *E_a_* and, consequently, the pyrolysis process could be described as a first-order reaction. From the TG curves, the conversion rate was relatively slow when the pyrolysis conversion rate of materials was between 0 and 10%, and in addition, Yao et al. [30] found that the reaction mechanism was altered under conditions of high conversion rates when studying the thermal decomposition of different natural fiber materials. As such, we focused on conversions ranging from 0.2 to 0.8 as opposed to the complete process. The correlation coefficients (*R^2^*) of kinetic parameters computed with the improved Coats-Redfern method with different conversion rates (α = 0.2–0.8) were greater than 0.9 (in Table 6), which indicates that the C-R method can better describe the pyrolysis process. The average *E_a_* value of WF was close to that of waste HDPE, at 182.7 and 178.9 kJ/mol, respectively. By comparing four kinds of WPCs with different ratios of wood flour, it can be found that, as the proportion of WF increased, the *E_a_* values also continued to increase. When the WF content accounted for 5 wt.% in the composites, the activation energy calculated by the C-R method was 115.8 kJ/mol, and when the wood flour content accounted for 20%, the *E_a_* value was 171.3 kJ/mol. This is because the pyrolysis carbonization of wood flour can form carbon on the plastic matrix in WPC materials, which prevents heat transfer, so that the reaction activation energy of the composite material increases.

### 3.3. Kinetics of WPC Reinforced with BFs

Activation energy trends of wood-plastic composites reinforced with basalt fibers are expressed in the plots of iso-conversional improved C-R and F-W-O methods in Figure 5. The F-W-O plot for B3 is listed in Figure 5a, and Figure 5b illustrates the plot of the modified C-R method. The fitted lines were nearly parallel as the conversion rate varied from 0.5 to 0.8 and the specific activation energy values are presented in Table 6. The average *E_a_* obtained by the modified C-R method of wood plastic composites loaded with 3.3 wt% BFs was relatively low at 145.5 kJ/mol, but the differences in average *E_a_* values with incremental BF contents became obvious. The average *E_a_* of BF/WF/HDPE composites increased to 204.6 kJ/mol after the fiber loading level increased to 6.7 wt.%. 

*E_a_* is a key factor used to ascertain the reaction rate, indicative of the minimum energy necessary to reach an activating molecule from the reactant molecule in the chemical reaction [42]. Different reactions are matched with different activation energies. Besides, a lower reaction *E_a_* results in more highly activated molecules and faster reaction rates at a given temperature. Conversely, a higher average *E_a_* results in a slower reaction rate. The modified Coats-Redfern, Flynne-Walle-Ozawa, and Friedman methods were adopted for ascertaining the *E_a_* of overall composite materials. The average *E_a_* values of WF/BF/HDPE composites calculated by modified Coats-Redfern and F-W-O methods were significantly greater than those of WF/HDPE composites. Moreover, the average *E_a_* increased, which indicates that the addition of BFs improves the thermal stability of the composite, increases the difficulty of thermal decomposition, and enhances the fire safety performance. This is because the BF is resistant to high temperatures, which enables the BF/WF/HDPE composites to have a better thermal stability from the surface and higher *E_a_*. Through the evaluation of the decomposition activation energy for wood-plastic composites and WF/BF/HDPE composites, it can be judged that composites coated with BFs enjoy prominent edges. However, it is worth mentioning that the *E_a_* values computed with the Friedman and Kissinger methods are quite different from those obtained by the other two methods, indicating that the Friedman and Kissinger models are not suitable for basalt fiber-reinforced wood-plastic composites. Scholars have made pyrolysis kinetic models of natural materials and fiber-reinforced polymer composites, and the *E_a_* values are consistent with different kinetic models [30,43,44,45]. The differences of *E_a_* in composite materials may be due to the mixture of basalt fibers and wood flour, which leads to changes in the pyrolysis process. In the initial stage of the pyrolysis process (*α* = 0.1–0.4), the accelerated decomposition process of hemicellulose, cellulose, and HPDE is prominent, which corresponds to the carbonization of cellulose and fracture of the HDPE backbone. Beyond *α* = 0.4, the fitting lines of materials were parallel and close. This is because the carbonization of wood flour in the initial stage hindered the transfer of heat, and basalt fibers still maintained a stable state at this stage, so the energy required for the pyrolysis reaction gradually increased, resulting in *E_a_* values higher than in the initial stage.

The linear relationship between logβ  and 1/T at different conversion rates obtained via the F-W-O method is displayed in Figure 4 and Figure 5. The specific results of *E_a_* and correlation coefficients are also listed in Table 6. The iso-conversional F-W-O method, defined as a specific integral method devoid of a functional model, can avert the trouble of choosing the reaction mechanism function until the activation energy is solved, without causing errors related to the hypothesis about the reaction mechanism function. Consequently, it is capable of examining the feasibility of the activation energy retrieved from the Coats-Redfern model method. Thus, the value of *E_a_* calculated by the F-W-O method was not significantly different from that computed with the modified Coats-Redfern method. The activation energies retrieved with the F-W-O method were relatively higher, but the variation in *E_a_* was the same as that obtained by the modified Coats-Redfern method. This change was similar to the literature [45,46], which proves that the kinetic parameters obtained by these two models are reliable. The activation energy calculated by the Friedman and Kissinger methods was quite different from the value obtained by the modified Coats-Redfern method, which shows that WPCs loaded with basalt fiber cannot be accurately modelled with the Friedman and Kissinger models.

Figure 6 shows the changes in activation energy in raw materials and composite materials with respect to the conversion rate. Figure 4a clearly shows that the *E_a_* values of wood flour and waste HDPE increase as the conversion rate increases, and the corresponding activation energy range (153–234 kJ/mol) of HDPE during the entire pyrolysis process is relatively narrow compared with that of wood flour (59–224 kJ/mol). Low *E_a_* with low rates of conversion and high *E_a_* with high rates of conversion of polymer composite materials (in Figure 6b, c) are presented in their pyrolysis processes, which imply different decomposition mechanisms [47]. It is obvious that, in Figure 6c, the *E_a_* values of WPCs reinforced with basalt fibers at any conversion rate over the full pyrolysis range are significantly higher than those of other WPCs without basalt fibers loading. These results demonstrate that polymer composites reinforced with basalt fibers show better thermal stability compared with other WPCs.

## 4. Conclusions

This research investigated how basalt fiber content in HDPE matrix composites affected the thermal decomposition process using dynamic thermogravimetric analysis. The thermal decomposition of WF/HDPE and BF/WF/HDPE composites at temperatures ranging between room temperature and 800 °C showed significant weight loss zones. The pyrolysis of cellulose/hemicellulose of wood flour occurred from 250 to 380 °C, and HDPE was mainly decomposed within a temperature range of 430–530 °C.Heating rates β have a great influence on the thermal weight loss of composites. With increased heating rates, the TG curves were significantly shifted to high temperatures. By eliminating the influence exerted by the heating rate on composite pyrolysis, the addition of BFs caused the decomposition temperature range, *T_0_*, and *T_P_* of the composite materials to shift towards a high temperature in comparison with wood-plastic composites, indicating that the introduction of BFs improved the thermal stability of composites.The modified Coats-Redfern, Flynn-Wall-Ozawa, Friedman, and Kissinger methods were used to model the activation energies of these materials. The results show the values of 115–171 kJ/mol for wood-plastic composites and 145–204 kJ/mol for other composites reinforced with basalt fibers obtained by the modified Coats-Redfern and Flynn-Wall-Ozawa methods throughout the pyrolysis processes, but the Friedman and Kissinger models were not suitable for polymer materials reinforced by mixed fillers. The addition of basalt fibers in polymer composites increased the activation energy and exhibited better stability during pyrolysis, and the increased degree of activation energy was significant as the content of basalt fibers increased.

## Figures and Tables

**Figure 1 polymers-12-02283-f001:**
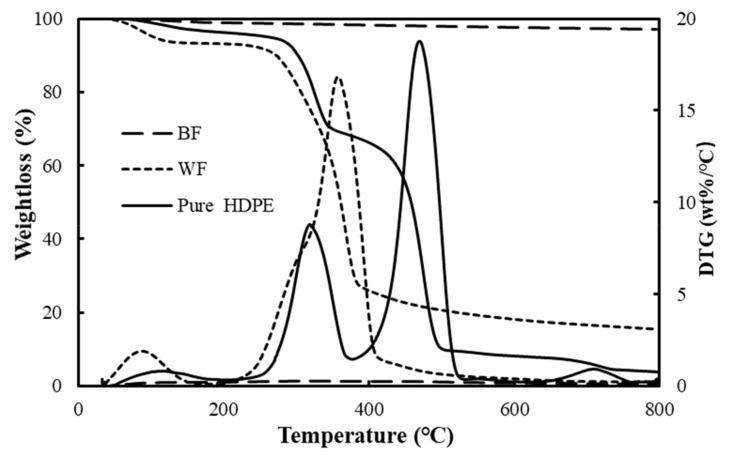
Thermal decomposition process of raw materials at a heating rate of 20 °C /min. BF, basalt fiber; WF, wood flour; HDPE, high density polyethylene.

**Figure 2 polymers-12-02283-f002:**
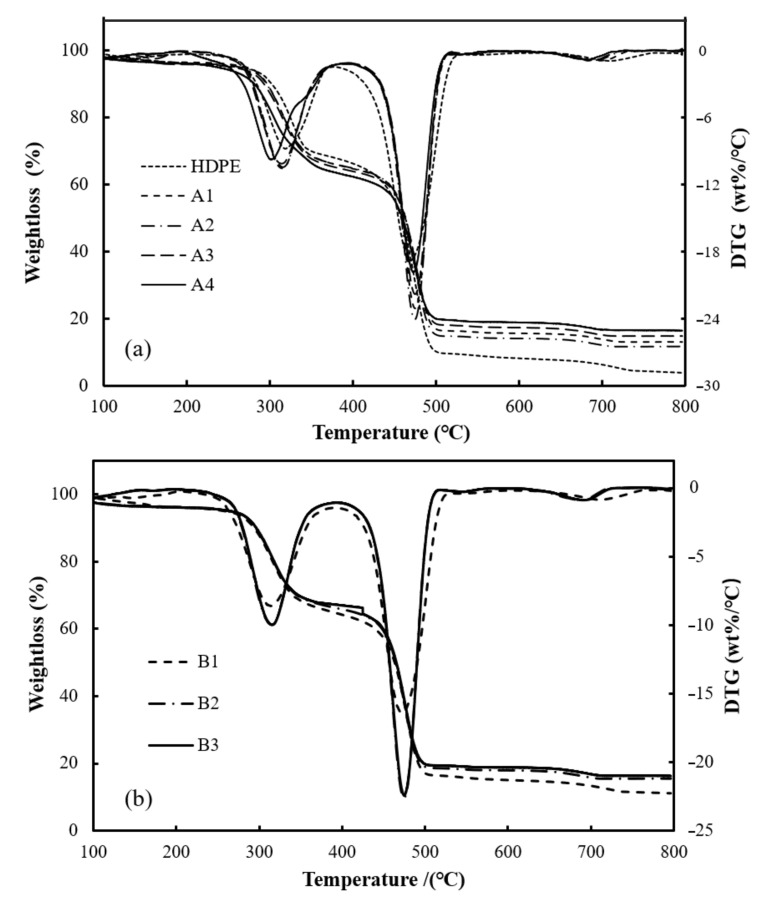
Thermal decomposition process of wood–plastic composites (WPCs) (**a**) and WPCs reinforced with BFs (**b**) at a heating rate of 20 °C/min.

**Figure 3 polymers-12-02283-f003:**
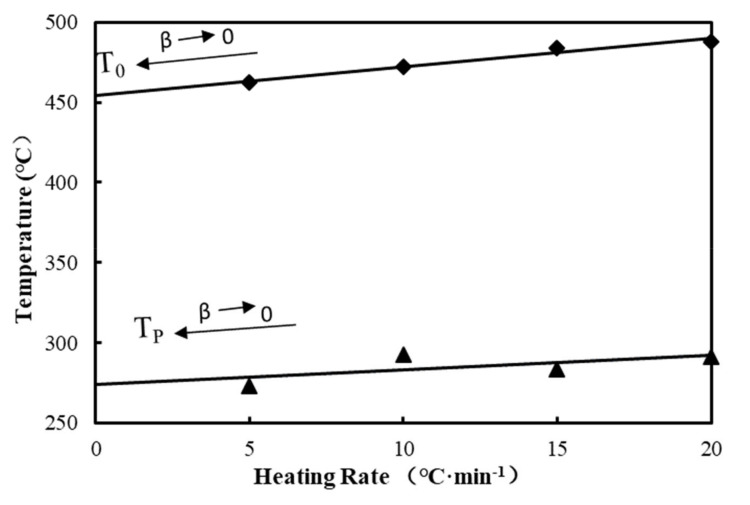
Determination of *T_0_* and T_P_ with B3 as an example.

**Figure 4 polymers-12-02283-f004:**
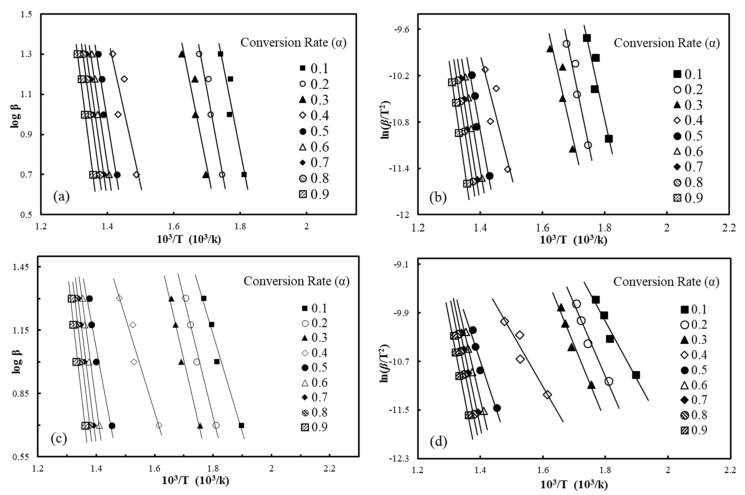
Iso-conversion plots of the iso-conversion plots of the Flynn-Wall-Ozawa (F-W-O) method for waste HDPE (**a**); A2 (**c**), the modified Coats-Redfern (C-R) method for waste HDPE (**b**); A2 (**d**).

**Figure 5 polymers-12-02283-f005:**
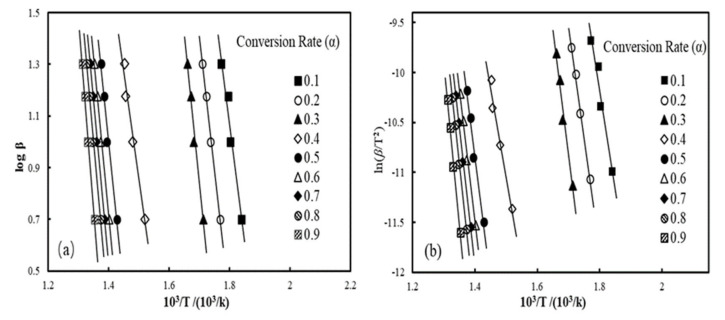
Iso-conversion plots of F-W-O method for B3 (**a**); example iso-conversion plots of the modified C-R method for B3 (**b**).

**Figure 6 polymers-12-02283-f006:**
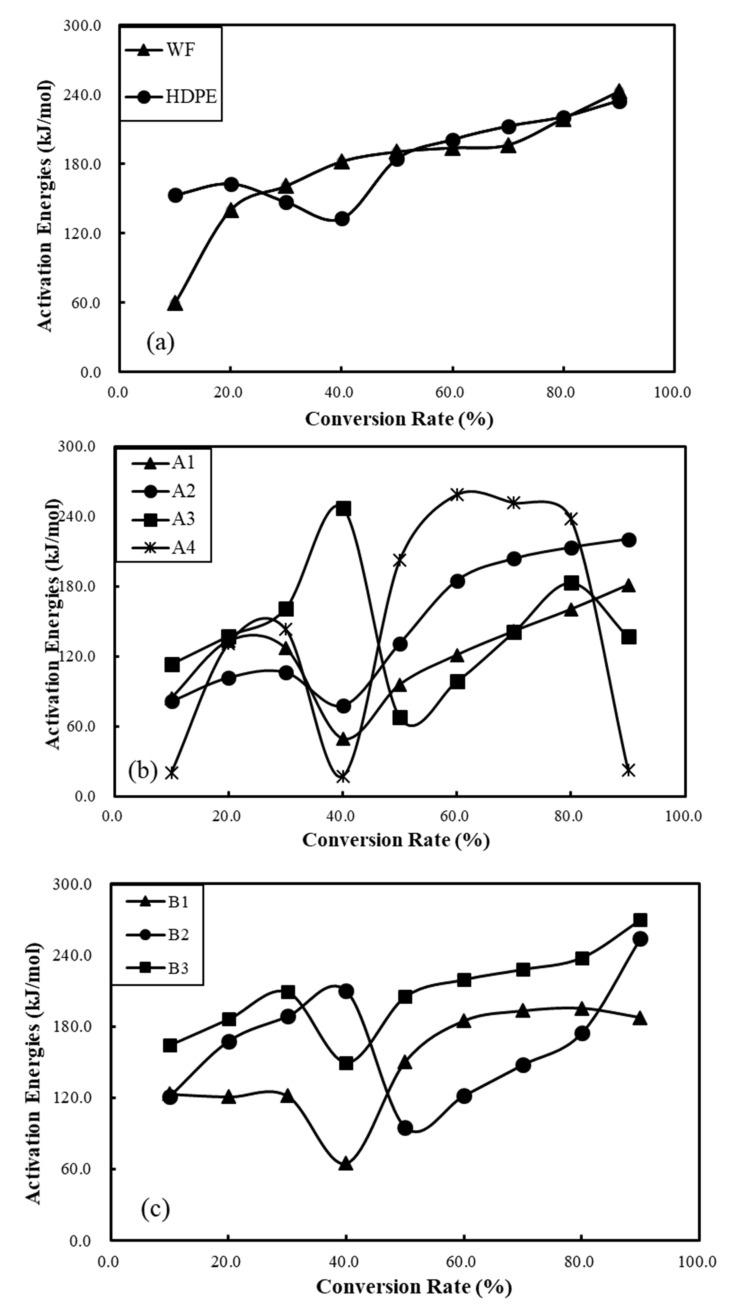
Effects of conversion rate on activation energy changes of raw materials (**a**), WPCs (**b**), and WPCs reinforced with BFs (**c**).

**Table 1 polymers-12-02283-t001:** Blending ratios of the composite materials (mass fraction). BF, basalt fiber; WF, wood flour; HDPE, high density polyethylene.

Samples	BF/WF	BF(%)	WF(%)	Coupling(%)	HDPE(%)
A0 (HDPE)	-	-	-	2	100
A1	-	-	5	2	95
A2	-	-	10	2	90
A3	-	-	15	2	85
A4	-	-	20	2	80
B1	1:2	3.3	6.7	2	90
B2	1:1	5	5	2	90
B3	2:1	6.7	3.3	2	90

**Table 2 polymers-12-02283-t002:** Kinetic methods for activation energy assessment in the research [31].

Method	Expression	Plots
Flynn-Wall-Ozawa	$\mathrm{lg}\beta =\frac{\mathrm{lg}\left(AE_{a}\right)}{Rf\left(a\right)}-2.315-\frac{0.4567E_{a}}{RT}$	lgβ vs. $\frac{1}{T}$
Coats-Redfern (modified)	$\ln\left[\frac{\beta }{T^{2}\left(1-2\frac{RT}{E_{a}}\right)}\right]=\ln\left[\frac{AR\left(1-\frac{2RT}{Ea}\right)}{\beta E_{a}}\right]-\frac{E_{a}}{RT}$	$\ln\left(\frac{\beta }{T^{2}}\right)$ vs. $\frac{1}{T}$
Friedman	$\ln\left(\frac{d\alpha }{dt}\right)=\ln\left[Af\left(\alpha \right)\right]-\frac{E_{a}}{RT}$	$\ln\left(\frac{d\alpha }{dt}\right)$ vs. $\frac{1}{T}$
Kissinger	$\ln\left(\frac{\beta }{T_{P}^{2}}\right)=\ln\left(\frac{AR}{E_{a}}\right)+\left(\frac{1}{T_{P}}\right)\left(-\frac{E_{a}}{R}\right)$	$\ln\left(\frac{\beta }{T_{P}^{2}}\right)$ vs. $\frac{1}{T_{P}}$

**Table 3 polymers-12-02283-t003:** Thermal decomposition parameters of raw materials.

Materials	β ^a^ (°C/min)	T_0_ (°C)	WL_0_ (%)	T_P_ (°C)	WL_P_ (%)	Residue (%)
BF	5	-	-	-	-	96.7
	10	-	-	-	-	96.8
	15	-	-	-	-	97.7
	20	-	-	-	-	97.1
WF	5	67.5	5	342.0	57.0	10.9
	10	76.5	5	353.2	55.8	13.7
	15	87.4	5	358.0	51.5	14.7
	20	103.8	5	357.4	46.2	15.5
HDPE	5	230.4	5	447.5	69.2	4.3
	10	253.4	5	463.1	66.1	5.7
	15	184.2	5	469.3	67.6	1.8
	20	262.3	5	469.6	62.2	3.9

a Capital: *β* = heating rate, *T* = temperature, *W* = weight loss, Residue = mean value of residual mass at different heating rates; subscript: o = onset, p = DTG peak.

**Table 4 polymers-12-02283-t004:** Thermal decomposition parameters of the composite materials.

Samples	β (°C/min)	T_0_ (°C)	WL_0_	T_P_ (°C)	WL_P_	Residue (%)
A1	5	224.7	5	442.4	70.8	8.5
	10	285.5	5	464.7	60.8	16.7
	15	190.7	5	469.1	66.4	8.7
	20	271.8	5	474.7	63.2	13.1
A2	5	207.9	5	449.3	64.3	15.4
	10	230.2	5	463.4	64.3	10.7
	15	168.7	5	469.1	67.4	7.5
	20	264.9	5	474.7	64.2	11.7
A3	5	203.7	5	447.5	67.3	5.8
	10	165.1	5	449.0	73.3	1.8
	15	190.7	5	469.1	66.4	8.7
	20	254.9	5	474.7	62.8	14.8
A4	5	231.3	5	449.7	65.3	13.1
	10	246.5	5	462.2	67.4	7.7
	15	166.0	5	469.1	65.9	11.7
	20	236.6	5	472.9	62.8	16.4
B1	5	236.1	5	448.7	62.4	17.1
	10	230.2	5	463.3	64.3	10.7
	15	257.9	5	470.6	63.4	10.6
	20	258.0	5	472.6	59.3	11.1
B2	5	148.6	5	440.4	67.5	10.6
	10	227.6	5	453.5	68.3	7.7
	15	198.4	5	469.9	63.7	12.5
	20	265.8	5	474.5	61.5	15.5
B3	5	219.7	5	448.4	65.1	9.2
	10	251.8	5	462.6	61.1	13.7
	15	198.3	5	469.6	63.8	12.3
	20	264.1	5	474.9	60.6	16.3

**Table 5 polymers-12-02283-t005:** Pyrolysis parameters (β → 0).

Samples	T_0_ ^a^ _β_ _→_ _0_ (℃)	WL_0_(%)	T_P β_ _→_ _0_ (℃)	WL_P_(%)	Residue ^b^ (%)
BF	-		-		-
WF	53.9	5	340.0	52.3(4.6)b	10.0
HDPE	226.0	5	444.3	66.3(3.0)	5.2
A1	231.6	5	437.4	65.3(4.3)	10.3
A2	190.6	5	443.6	65.0(1.5)	14.9
A3	158.8	5	434.7	67.4(4.4)	0.8
A4	236.2	5	444.3	65.3(1.9)	8.7
B1	222.2	5	444.1	62.4(2.2)	16.9
B2	129.5	5	429.8	65.2(3.2)	6.7
B3	213.6	5	442.3	62.7(2.1)	7.9

„^a^
*T_0_* = initial decomposition temperature when *β*→0, *T_P_* = maximum decomposition temperature when *β* → 0 “;„ ^b^ Values from materials with mean value and standard deviation“.

**Table 6 polymers-12-02283-t006:** Coats-Redfern (C-R) and Flynn-Wall-Ozawa (F-W-O) assessments of the apparent activation energy of the materials.

	Coats-Redfern		Flynn-Wall-Ozawa		Friedman			Kissinger	
	E_a_(kJ·mol^−1^)	R^2^	E_a_(kJ·mol^−1^)	R^2^	E_a_(kJ·mol^−1^)	R^2^	E_a_(kJ·mol^−1^)		R^2^
BF	-	-	-	-	-	-	-		-
WF	182.7 (24.2)	0.98 (0.03)	183.3 (23.4)	0.99 (0.04)	15.8 (41.0)	0.26 (0.32)	241.1		0.91
HDPE	178.9 (31.5)	0.91 (0.09)	181.0 (30.7)	0.95 (0.05)	3.0 (20.9)	0.10 (0.10)	239.0		0.93
A1	115.8 (34.3)	0.74 (0.27)	120.8 (32.7)	0.76 (0.25)	28.8 (25.6)	0.71 (0.45)	171.7		0.95
A2	143.8 (51.3)	0.96 (0.02)	147.4 (49.6)	0.97 (0.02)	14.8 (49.1)	0.81 (0.19)	234.1		0.99
A3	146.3 (55.3)	0.71 (0.23)	149.7 (52.3)	0.90 (0.15)	38.0 (53.1)	0.45 (0.26)	162.5		0.79
A4	171.3 (92.4)	0.61 (0.36)	173.4 (88.6)	0.62 (0.35)	37.7 (43.7)	0.21 (0.15)	250.5		1.00
B1	145.5 (45.4)	0.97 (0.03)	149.0 (43.9)	0.99 (0.01)	4.8 (31.5)	0.66 (0.22)	233.2		0.97
B2	156.2 (37.9)	0.95 (0.02)	159.1 (35.5)	0.96 (0.01)	10.2 (83.9)	0.62 (0.43)	156.1		0.97
B3	204.6 (27.9)	0.98 (0.01)	205.3 (27.0)	0.99 (0.01)	8.1 (18.8)	0.55 (0.45)	222.5		1.00

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
