# Peer review of "Thermal Decomposition Kinetics of Basalt Fiber-Reinforced Wood Polymer Composites"

_polymers, 2020, doi:10.3390/polym12102283_

Round 1
Reviewer 1 Report
The article “Thermal Decomposition kinetics of Basalt Fiber Reinforced Wood Polymer Composites” by Zhang and Huang presents an interesting subject of the kinetic analysis of the thermodegradation of polyethylene-based hybrid composites. The problem of manufacturing hybrid composites containing both natural fillers as well as reinforcement in the form of basalt fibers is an increasingly discussed research issue and finds an important place in industrial practice. Therefore, in-depth studies on the effects of both fillers on the phenomena of thermal decomposition are warranted, which may be an important source of information in future studies (e.g. on flammability and fire safety). In my opinion, the performed calculation procedures do not raise any major objections, but the article, before it can be accepted for publication, should be supplemented and corrected in accordance with the comments presented below.
- The authors wrote that BF hardly decomposed under 800 ° C, which is not true. Please correct this wording accordingly.
- In the introduction part, the authors should clearly emphasize and describe the disadvantages associated with lowering thermal stability that characterize WPC composites based on HDPE matrix. This should be justified by presenting literature data and clearly demonstrating what changes can be expected by using two fillers simultaneously, taking into account natural fibers (wood flour) and basalt fibers. The article describes a detailed case of the analysis of the thermal properties of HDPE-WF-BF composites, therefore the presented cases of previous literature reports should refer in particular to changes in thermal properties and flammability of similar hybrid materials.
- Please provide justification for the selection of material compositions. In fact, it is difficult to refer to the presented materials when the volumetric filler fractions are not shown and the material series containing unequal amounts of additives subject to different temperature loads are compared with each other. I would like authors for suitable comments.
- The mass of samples should be indicated in the test methodology.
- Please check the article for typos, incl. resulting from auto-correction (e.g. Thermos-gravimetric)
- Line 266. Please specify the statement regarding the change in the pyrolysis process.
- Please provide a more detailed description of the phenomena occurring during the degradation of the lignocellulosic filler. In addition, one should refer to the chemical composition of the filler (including the content of individual components) and, based on the literature analysis, propose possible mechanisms of changes that may occur as a result of introducing basalt fibers.
Author Response
Dear editor and reviewers:
Thank you very much for your time and effort in reviewing our manuscript. We sincerely appreciate your suggestions, which have helped to greatly improve the manuscript. According to your comments, we have revised the manuscript as follows:
Responds to Reviewer #1:
Comment 1: The authors wrote that BF hardly decomposed under 800 °C, which is not true. Please correct this wording accordingly.
Respond: We have changed the description of the decomposition of basalt fibers in the manuscript. In fact, Hao et al. (1) compared the thermal stability of glass fibers and basalt fibers and found the mass loss of glass fibers was rather small at 1.76%, and the mass loss of basalt fibers was even less at 0.74%, according to TGA at 900 °C. The main components of basalt fibers are SiO2, Al2O3, FeO+Fe2O3, CaO, MgO, and TiO2, which are typically stable compounds. A small amount of FeO will undergo a reduction to form Fe2O3, but most of the compounds remain at steady state during the heating period.
Comment 2: In the introduction part, the authors should clearly emphasize and describe the disadvantages associated with lowering thermal stability that characterize WPC composites based on HDPE matrix. This should be justified by presenting literature data and clearly demonstrating what changes can be expected by using two fillers simultaneously, taking into account natural fibers (wood flour) and basalt fibers. The article describes a detailed case of the analysis of the thermal properties of HDPE-WF-BF composites, therefore the presented cases of previous literature reports should refer in particular to changes in thermal properties and flammability of similar hybrid materials.
Respond: We have added this to the introduction and provided relevant literature reports to illustrate the effect of fillers on thermal stability.
Comment 3: Please provide justification for the selection of material compositions. In fact, it is difficult to refer to the presented materials when the volumetric filler fractions are not shown and the material series containing unequal amounts of additives subject to different temperature loads are compared with each other. I would like authors for suitable comments.
Respond: Waste HDPE is difficult to be decomposed in a short time in the natural environment, and wood powders are usually treated by traditional incineration. Therefore, the use of waste HDPE and wood powders is not only conducive to environmental protection, but also to resource recycling. Basalt fiber was selected because of its excellent properties such as high strength, corrosion resistance, high temperature resistance, and direct environmental degradation.
All composites are extruded under the same temperature load. Table 1 has also given the mass fraction of each material. First, pure HDPE is compared with wood plastic composites, and then basalt fibers are added into WPCs to explore the influence of the ratio of basalt fibers to wood flour on the composite materials. In the BF/WF/HDPE composites, the mass fraction of HDPE is 90%, which can be compared with A2 in the table (HDPE is also 90 wt%) for comparison.
Comment 4: The mass of samples should be indicated in the test methodology.
Respond: The sample quality has been indicated in the test method.
Comment 5: Please check the article for typos, incl. resulting from auto-correction (e.g. Thermos-gravimetric)
Respond: We have thoroughly checked the whole manuscript and corrected the grammar and spelling mistakes according to the reviewer’s comments, and we have further polished the language.
Comment 6: Line 266. Please specify the statement regarding changes in the pyrolysis process.
Respond: We have a detailed description, and we thank the reviewer for these comments and suggestions.
Comment 7: Please provide a more of the phenomena occurring during the degradation of the lignocellulosic filler. In addition, one should refer to the chemical composition of the filler (including the content of individual components) and, based on the literature analysis, propose possible mechanisms of changes that may occur as a result of introducing basalt fibers.
Respond: The detailed description in the pyrolysis process of lignocellulose fillers has been described in detail at line 154, and the possible reactions of basalt fibers in the pyrolysis process of the entire composite materials have been discussed at line 207.
Reviewer 2 Report
The paper presents a green material which can be used in many useful purposes. The following points should count to improve the manuscript.
- More discussion need to clearly understand the PE, PP, and HDPE material in introduction. Their advantage and disadvantage in different application should describe.
- The main highlighting of the paper is TGA analysis. They use 4 different rates in TGA analysis. They used mixed nitrogen and oxygen during TGA analysis. Please add at least one Figure to compare the different rate. Why you chose mixed gas combination.
- FT-IR analysis of the melt compounded of BF, WF, HDPE and 2% MAPE will be improved the manuscript.
Author Response
Dear editor and reviewers:
Thank you very much for your time and effort in reviewing our manuscript. We sincerely appreciate your suggestions, which have helped to greatly improve the manuscript. According to your comments, we have revised the manuscript as follows:
Responds to Reviewer #2:
Comment 1: More discussion need to clearly understand the PE, PP, and HDPE material in introduction. Their advantage and disadvantage in different application should describe.
Respond: Compared with PP, HDPE has a better low temperature resistance and lower costs, and in contrast with LDPE, HDPE has mechanical property advantages of higher rigidity, hardness, and tensile strength. This discussion regarding advantages of HDPE has been included in the introduction.
Comment 2: The main highlighting of the paper is TGA analysis. They use 4 different rates in TGA analysis. They used mixed nitrogen and oxygen during TGA analysis. Please add at least one Figure to compare the different rate. Why you chose mixed gas combination.
Respond: Nitrogen and oxygen were used for pyrolysis analysis in references (2, 3), and we have adopted these conditions. A small amount of oxygen is usually added to pyrolysis kinetics to enable a more efficient pyrolysis reaction.
Comment 3: FT-IR analysis of the melt compounded of BF, WF, HDPE and 2% MAPE will be improved the manuscript.
Respond: We understand that FT-IR analysis of material may better reveal the functional groups between materials and help explain the bond connections between the basalt fiber-reinforced WPCs. However, in the present study, we mainly focus on thermal decomposition kinetics, and we think that TG analysis alone may not be optimal, but should be sufficient to draw the conclusion that basalt fibers could improve the average Ea and further enhance the thermal stability and flame resistance of composites.
The reviewer is thanked for this suggestion,FT-IR analysis was reported in the another manuscript, “Mechanical , morphology and thermal expansion properties of basalt fiber- reinforced wood polymer composites” , what is being reviewed.
Round 2
Reviewer 1 Report
The authors correct the manuscript according to mentioned in former review comments. Before publication, some minor editorial issues should be corrected.
- The quality of the equations and figures is insufficient. They probably were converted into the images during the changing of word file format or pdf build. Please correct this.
- Some typos occurred in the prepared text. The authors should carefully read the final manuscript version. In the future, I recommend avoiding resending the revised manuscript in ”revision mode”, it is really hard to read for the reader who is not an author and the whole manuscript become illegible.
Author Response
Dear editor and reviewers:
Thank you very much for your time and effort in reviewing our manuscript. We sincerely appreciate your suggestions, which have helped to greatly improve the manuscript. According to your comments, we have revised the manuscript as follows:
Responds to Reviewer #1:
Comment 1: The quality of the equations and figures is insufficient. They probably were converted into the images during the changing of word file format or pdf build. Please correct this.
Respond: Thank you very much for raising the problem that we did not notice before. We have edited and corrected the relevant equations in this paper.
Comment 2: Some typos occurred in the prepared text. The authors should carefully read the final manuscript version. In the future, I recommend avoiding resending the revised manuscript in “revision mode”, it is really hard to read for the reader who is not an author and the whole.
Respond: We have carefully read the final manuscript and corrected some misspellings once again. Thank you again for your careful reading of this paper.
Reviewer 2 Report
The manuscript has been revised accordingly. Please add your FT-IR ref (you mentioned submitted to somewhere) in experimental 2.2, interested reader can read that one.
Author Response
Dear editor and reviewers:
Thank you very much for your time and effort in reviewing our manuscript. We sincerely appreciate your suggestions, which have helped to greatly improve the manuscript. According to your comments, we have revised the manuscript as follows:
Responds to Reviewer #2:
Comment 1: The manuscript has been revised accordingly. Please add your FT-IR ref (you mentioned submitted to somewhere) in experimental 2.2, interested reader can read that one.
Respond: The reviewer is thanked for this suggestion,We also want to refer to the FTIR reference in this paper. But the manuscript include the FT-IR is still being reviewed(round 2) in the “Journal of Forestry Engineering”, the manuscript without DOI is not available currently for cited. If another manuscript will get the DOI before this paper published, we will cited this reference as soon as possible.We are very regretful for your efforts.